

# EuLerian Identification of Ascending air Streams (ELIAS 2.0) in Numerical Weather Prediction and Climate Models. Part II: Model application to different data sets

Julian F. Quinting[1], Christian M. Grams[1], Annika Oertel[1], and Moritz Pickl[1]

[1]Institute of Meteorology and Climate Research (IMK-TRO), Karlsruhe Institute of Technology, Germany

**Correspondence:** Julian F. Quinting (julian.quinting@kit.edu)

**Abstract.** Warm conveyor belts (WCBs) affect the atmospheric dynamics in midlatitudes and are highly relevant for total and extreme precipitation in many parts of the extratropics. Thus, these air streams and their effect on midlatitude weather should be well represented in numerical weather prediction (NWP) and climate models. This study applies newly developed convolutional neural network (CNN) models which allow the identification of footprints of WCB inflow, ascent, and outflow from a limited

number of predictor fields at comparably low spatio-temporal resolution. The goal of the study is to demonstrate the versatile applicability of the CNN models to different data sets and that their application yields qualitatively and quantitatively similar results as their trajectory-based counterpart which is most frequently used to objectively identify WCBs but requires data at higher spatio-temporal resolution which is often not available and is computationally more expensive. First, an application to reanalyses reveals that the well-known relationship between WCB ascent and extratropical cyclones as well as between WCB

outflow and blocking anticyclones is also found for WCB footprints identified with the CNN models. Second, the application to Japanese 55-year reanalyses shows how the CNN models may be used to identify erroneous predictor fields that deteriorate the models' reliability. Third, a verification of WCBs in operational European Centre for Medium-Range Weather Forecasts (ECMWF) ensemble forecasts for three Northern Hemisphere winters reveals systematic biases over the North Atlantic with both the trajectory-based approach and the CNN models. The ensemble forecasts' skill tends to be lower when being evaluated

with the trajectory approach due to the fine-scale structure of WCB footprints in comparison to the rather smooth CNN-based WCB footprints. A final example demonstrates the applicability of the CNN models to a convection permitting simulation with the ICOsahedral Nonhydrostatic (ICON) NWP model. Our study illustrates that deep learning methods can be used efficiently to support process-oriented understanding of forecast error and model biases, and opens numerous directions for future research.

## 1 Introduction

Extratropical cyclones are accompanied by coherent air streams which ascend cross-isentropically from the lower to the upper troposphere within two days – so-called warm conveyor belts (WCBs; Browning et al., 1973; Harrold, 1973; Carlson, 1980). WCBs form an integral part of the extratropical atmospheric circulation as they release large amounts of latent heat, are





responsible for the major part of precipitation associated with extratropical cyclones, and are the primary cloud producing
extratropical flow structure (e.g., Browning, 1985; Eckhardt et al., 2004; Pfahl et al., 2014; Joos, 2019).

Typically, WCBs originate in the marine boundary layer of an extratropical cyclone's warm sector and ascend poleward along the cyclone's cold front (Wernli and Davies, 1997). This WCB ascent, which can be slantwise or convective in nature (e.g., Neiman and Shapiro, 1993; Rasp et al., 2016; Oertel et al., 2019), is accompanied by latent heat release on the order of 20 K due to phase changes during cloud formation (Eckhardt et al., 2004; Madonna et al., 2014). The latent heat release
subsequently affects the production and destruction of potential vorticity (PV) due to vertical gradients of the diabatic heating rates (Hoskins et al., 2007). Assuming a positive vertical component of absolute vorticity, cyclonic PV is generated below and close to the level of maximum heating (e.g., Stoelinga, 1996; Wernli and Davies, 1997). If the WCB ascent occurs close to the center of an extratropical cyclone, the lower- to mid-tropospheric cyclonic PV contributes to the cyclone's PV tower and the WCB is thus important for its evolution and intensity (e.g., Rossa et al., 2000; Binder et al., 2016). Accordingly,
extratropical cyclones and WCB ascent are directly linked. Above the level of maximum heating, PV is destroyed such that the net PV change from the WCB inflow to the WCB outflow is approximately zero (Methven, 2015). Still, the latent heat release during the WCB ascent leads to a net cross-isentropic transport of lower-tropospheric low-PV air into the upper troposphere where it contributes along with its diabatically amplified divergent outflow to the formation of anticyclonic PV anomalies (e.g., Pomroy and Thorpe, 2000; Ahmadi-Givi et al., 2004; Grams et al., 2011; Bosart et al., 2017). These anticyclonic PV anomalies
may trigger or modify downstream Rossby waves (Röthlisberger et al., 2018) or may contribute to the onset and maintenance of blocking anticyclones (e.g., Pfahl et al., 2015; Grams and Archambault, 2016; Steinfeld and Pfahl, 2019). For example, Steinfeld and Pfahl (2019) found that almost 10% of air masses in blocking anticyclones had ascended in WCBs during the 7 days before reaching the blocking region.

This brief overview highlights that systematic forecast errors associated with WCBs may project on the representation of
the extratropical atmospheric circulation in NWP and climate models. Indeed, significant biases exist in state-of-the-art NWP systems concerning the climatological occurrence frequency of WCB inflow, ascent, and outflow (Wandel et al., 2021). The systematic verification of the three WCB stages in an ensemble NWP system was made possible by a novel statistical approach that allows the identification of WCB footprints without the need to perform trajectory calculations (Quinting and Grams, 2021a). Though in their study the identification of WCBs was based on a logistic regression approach, a refined statistical
approach based on convolutional neural network (CNN) models is introduced in a companion study (Quinting and Grams, 2021b, hereafter referred to as Part I). The study at hand aims to show different applications of the CNN models and to demonstrate that the CNN models yield qualitatively and quantitatively similar results as their Lagrangian counterpart which is computationally more expensive and requires data at higher spatio-temporal resolution. The exemplary applications are

1.  a climatological investigation of the relationship between footprints of WCB ascent and extratropical cyclones as well
as between footprints of WCB outflow and blocking anticyclones,

2.  a comparison of the CNN models' reliability when being applied to a reanalysis data set other than the one it was trained on and its usefulness to identify differences in the predictor variables provided to the CNN,





3. an evaluation of operational ECMWF ensemble WCB forecasts in terms of their biases and skill,

4. and a case study of a WCB in a convection permitting simulation performed with the ICOsahedral Nonhydrostatic NWP
model (ICON Zängl et al., 2015) which is independent from the ECMWF data the CNN was trained on.

The various data sets used in this study are introduced in Section 2. The comparison of the CNN-based and trajectory-based diagnostics is presented in Section 3. Directions for future research are outlined together with a concluding discussion in Section 4.

## 2   Data and Methods

### 2.1   ERA-Interim data

The ECMWF's Interim reanalyses (ERA-Interim; Dee et al., 2011) form the basis for the climatological investigations of the relationship between footprints of WCBs identified with the trajectory and CNN approach, extratropical cyclones, and blocking anticyclones. The reanalysis data are derived 6-hourly at 00, 06, 12, 18 UTC and are remapped from its original T255 spectral resolution to a regular $1° \times 1°$ latitude-longitude grid. As in Part I, the testing period December, January, February (DJF) from
01 January 2005 to 31 December 2016 is chosen for all climatological analyses presented in this study.

#### 2.1.1   Trajectory-based WCB climatology

For the trajectory-based WCB climatology by Madonna et al. (2014), 48-h kinematic forward trajectories are calculated using the horizontal and vertical wind components on all available model levels in ERA-Interim with the LAGRangian ANalysis TOol (LAGRANTO; Wernli and Davies, 1997; Sprenger and Wernli, 2015). The initial starting points of WCBs are found
by seeding trajectories from a global $80\,\mathrm{km} \times 80\,\mathrm{km}$ equidistant grid in the horizontal and vertically every $20\,\mathrm{hPa}$ from 1050 to $790\,\mathrm{hPa}$. After calculating the forward trajectories from all starting points, only those trajectories are kept as WCBs which ascend at least by $600\,\mathrm{hPa}$ in $48\,\mathrm{h}$ and which are matched with an extratropical cyclone mask (Wernli and Schwierz, 2006) at least once during the 48-h period. We define WCB inflow, WCB ascent, and WCB outflow by binning all identified WCB parcel locations at a given time into three vertical layers (Schäfler et al., 2014). WCB inflow refers to those air parcels being
located below $800\,\mathrm{hPa}$, ascent includes those between 800 and $400\,\mathrm{hPa}$, and outflow refers to all air parcels above $400\,\mathrm{hPa}$. For the three layers, air parcel locations are gridded on a regular $1° \times 1°$ latitude-grid, i.e., grid points without/with a WCB air parcel are labelled as 0/1 yielding two-dimensional binary footprints for each of the three WCB stages.

#### 2.1.2   CNN-based WCB climatology

The CNN-based WCB climatology is taken from Part I which provides a detailed description of the underlying models. In
short, for each of the three WCB stages of WCB inflow, ascent, and outflow separate CNN models with variants of the UNet architecture (Ronneberger et al., 2015) are implemented. This architecture was originally designed to perform image segmentation tasks using the RGB-values of images as predictors. In this study, the predictors for each CNN model are in total five





predictors, four of which are meteorological parameters derived from temperature, geopotential height, specific humidity and the horizontal wind components at the 1000, 925, 850, 700, 500, 300, and 200 hPa isobaric surfaces (see Table 1 in Part I). For WCB ascent, a fifth predictor is the 30-day running mean trajectory-based climatological WCB occurrence frequency. Taking into account the time-lag between the WCB stages, a fifth predictor for WCB inflow is the conditional probability of WCB ascent predicted by the CNN 24 hours later than the corresponding WCB inflow time. Conversely, the fifth predictor for WCB outflow is the conditional probability of WCB ascent 24 hours earlier than the corresponding WCB outflow time. As for the trajectory-based data, the conditional probabilities predicted by the CNN models are converted to two-dimensional binary footprints of WCB inflow, ascent, and outflow on a regular $1° \times 1°$ latitude-grid by applying grid-point specific decision thresholds.

### 2.1.3 Extratropical cyclone data

In the trajectory-based WCB climatology only those rapidly ascending air streams are considered as WCBs that occur in the vicinity of extratropical cyclones. Madonna et al. (2014) account for this relationship by keeping only those rapidly ascending trajectories as WCBs which are matched at least once during their 48-h life time with an extratropical cyclone mask (Wernli and Schwierz, 2006). The CNN models of Part I do not use the extratropical cyclone mask information as predictor such that it is not upfront clear whether the CNN models correctly reproduce the relationship between WCBs and extratropical cyclones. Here, we test for this relationship by matching the trajectory-based and CNN-based masks of WCB ascent with the extratropical cyclone masks provided by the climatological data set of Sprenger et al. (2017). In their data set, which is based on the same ERA-Interim data introduced in Section 2.1, extratropical cyclone masks include all regions delimited by the outermost closed sea level pressure contour enclosing one or several local sea level pressure minima (Wernli and Schwierz, 2006). Objects of WCB ascent are chosen since this stage of the WCB life cycle occurs closest to the center of extratropical cyclones. In a first step, all CNN-based and trajectory-based footprints of WCB ascent at a certain time step are assigned with an identifying number. We then check for each identified footprint whether at least one grid point is collocated with the mask of an extratropical cyclone. If this criterion is fulfilled the entire WCB footprint is considered to be matched with an extratropical cyclone. The climatological matching frequency is then the ratio of matched WCB footprints against all (matched and non-matched) footprints.

### 2.1.4 Blocking anticyclone data

As outlined in the introduction, the WCB outflow may be directed into upper-tropospheric blocking anticyclones. We test whether this relationship can be reproduced with the CNN-based WCB diagnostic by matching masks of blocks (Pfahl et al., 2015; Sprenger et al., 2017) with masks of WCB outflow identified with the trajectory approach and CNN the models. Following the definition of Schwierz et al. (2004) and Croci-Maspoli et al. (2007), blocks in the Northern Hemisphere are defined as regions where the negative anomaly of vertically averaged PV between 150 and 500 hPa is less than –1.3 PVU (1 PVU = $10^{-6}$ K kg$^{-1}$ m$^2$ s$^{-1}$) and persists for at least 5 days. The anomalies of vertically averaged PV are calculated as deviations from the monthly climatology of vertically averaged PV.



## 2.2 JRA-55 data

In order to test the sensitivity of the CNN models to the input data, we apply the models for the testing period (01 January 2005 to 31 December 2016) to the Japanese 55-year reanalysis data (JRA55 Kobayashi et al., 2015; Harada et al., 2016). The data are derived with the same temporal resolution and at the same pressure levels as the ERA-Interim data. Further, the data
are remapped from their native T319 resolution to a regular $1° \times 1°$ grid spacing.

## 2.3 Operational ECMWF IFS ensemble forecasts

We evaluate ECMWF's operational IFS ensemble forecasts (ECMWF, 2020) for the period 01 December to 28 February in the three Northern Hemisphere winters 2018/19, 2019/20, 2020/21 over the North Atlantic region (90°W to 30°E; 15° to 80°N) with the trajectory-based and the CNN-based approach. Here we combine the three model cycles CY45r1, CY46r1,
and CY47r1 without considering differences between the three model versions. The ensemble forecast consists of one control forecast and 50 perturbed forecasts which are initialized twice-daily at 00 and 12 UTC. Though forecasts are run up to 15 days lead time, we restrict our analysis for computational reasons to forecast lead times up to 144 h (6 days) at 6-hourly time steps. All forecasts are remapped from their original resolution of TCo639 to a regular $1° \times 1°$ latitude-longitude grid. For the trajectory calculations, forecasts were derived on model levels 39 to 91 (surface to about 16 km) in near real-time since these
data are not archived in the long-term Meteorological Archival and Retrieval System (MARS) at ECMWF. Further, the data required for the trajectory calculations were derived for a smaller region extending from 130°W to 80°E and from 15°N to 80°N due to data storage capacities. The trajectory calculations follow the initial setup for WCB ensemble forecasts of (Schäfler et al., 2014): 48-h forward trajectories are started from a $100\,km \times 100\,km$ equidistant grid in the horizontal and vertically every 50 hPa from 1000 to 700 hPa. Compared to the original climatological data set (cf. Section 2.1.1) fewer starting levels are
chosen for computational reasons. For forecast lead times of less than 48 h, the trajectories are calculated from a combination of short-range forecasts at 0 and 6 h lead time obtained from earlier initialization times and the actual forecast. After calculating the forward trajectories from all starting points, only those trajectories are kept as WCBs which ascend by at least 600 hPa in 48 h. A matching with an extratropical cyclone mask as in the original Lagrangian definition (Madonna et al., 2014) was not performed. For the CNN-based WCB identification, ensemble forecast data were downloaded globally and on pressure levels
specified in Part I.

Though data on a global grid are required to apply the CNN models, the reduced number of vertical pressure levels compared to the large number of model levels reduces the needed disk space by roughly one third. A single ensemble forecast needed for the trajectory calculation described above roughly amounts to 10.9 Gigabyte in the General Regularly-distributed Information in Binary form (GRIB) format. In contrast, the forecast data needed for the CNN-based diagnostic only amounts to 7.2 Gigabyte
in GRIB format. Most importantly the computational time needed for the two diagnostics differs considerably. The calculation and gridding of the trajectories for a single ensemble forecast takes roughly 14 hours on a single CPU at 3.60 GHz. In contrast, it takes roughly 20 minutes on the same CPU to process one ensemble forecast with the CNN models which corresponds to a 40-fold reduction in computing time.





We evaluate the operational ensemble forecasts in terms of the mean error (hereafter referred to as *bias*) in WCB inflow,
ascent, and outflow frequency compared to the short-range control forecasts at 0 to 6 h lead time (hereafter referred to as
pseudo-analysis). Footprints identified with the CNN models (trajectory approach) in the ensemble forecast are verified against
footprints identified with the CNN models (trajectory approach) in the pseudo-analysis. The mean error as function of grid
point $x_i$ and time $t$ is defined as

$$ME(x_i,t) = \frac{1}{N} \sum_{n=1}^{N} (y_n(x_i,t) - o_n(x_i,t)) \tag{1}$$

with $0 \leq y_n(x_i,t) \leq 1$ being the ensemble mean WCB frequency at a specific grid point and forecast lead time and $o_n(x_i,t)$ the
corresponding dichotomous observation from the pseudo-analysis. $N$ denotes the number of forecasts that are used to calculate
the mean error ($N$=540). The forecast skill of the ensemble forecast system is evaluated with the Brier Skill Score (BSS)

$$BSS(x_i,t) = 1 - \frac{BS(x_i,t)}{BS_{CLIM}(x_i,t)} \tag{2}$$

which compares the Brier Score of the ensemble forecast system

$$BS(x_i,t) = \frac{1}{N} \sum_{n=1}^{N} (y_n(x_i,t) - o_n(x_i,t))^2 \tag{3}$$

against the Brier Score of a climatological reference forecast

$$BS_{CLIM}(x_i,t) = \frac{1}{N} \sum_{n=1}^{N} (\overline{o_n}(x_i,t) - o_n(x_i,t))^2 \quad . \tag{4}$$

Here, the seasonal climatology $\overline{o_n}(x_i,t)$ is calculated from the pseudo-analysis for the period 01 December to 28 February in
the winters 2018/19, 2019/20, 2020/21. The ensemble forecast system performs better than the reference forecast for a BSS
larger than 0 and is perfect at a BSS of 1.

## 2.4 ICON data

To simulate a WCB case study in convection permitting resolution, we run the non-hydrostatic model ICON (ICOsahedral
Nonhydrostatic; Zängl et al., 2015) globally with the operational resolution of approximately 13 km (R03B07) with 90 verti-
cal levels between the surface and 23 km height, and include two refined nests with resolutions of 6.5 km (R03B08) and 3.3 km
(R03B09), respectively, that focus on the WCB ascent region and are coupled with a two-way feedback. The simulation is ini-
tialized with the operational ECMWF IFS analysis at 00 UTC 03 October 2016 and run for 5 days with a time step of dt = 120 s
in the global domain (corresponding to dt = 60 s and dt = 30 s, in the respective nests). We apply the two-moment microphysics
scheme (Seifert and Beheng, 2006) with 6 prognostic hydrometeor types (cloud and rain droplets, ice, snow, graupel, and hail).
Deep convection in the global domain is parameterized with a Tiedtke-Bechtold scheme (Bechtold et al., 2008; Tiedtke, 1989),
while in both refined nests deep convection is treated explicitly and only shallow convection is parameterized.

The global nested simulation setup allows to compute trajectories with a high resolution in the refined nests, while the
coupling to the global domain ensures that the nests and the global domain stay closely together during the 5 day of WCB





simulation. This is important because global output is required for the application of the CNN models. Hence, this setup allows to directly compare the performance of the CNN models in high-resolution ICON simulations and the direct comparison with
185 high-resolution WCB trajectories.

Trajectories are calculated with LAGRANTO based on hourly horizontal and vertical wind fields from the larger nest with a resolution of 6.5 km which are interpolated to a regular 0.1°×0.1° grid on the original 90 vertical model levels. For the WCB case study, 48-h forward trajectories are started every hour and every 25 km within 60°W/35°N and 10°W/55°N from 7 vertical levels (250, 500, 750, 1000, 1500, 2000, 2500 m above ground). WCB trajectories are subsequently selected as
trajectories with an ascent rate of at least 600 hPa in 48 h. In contrast to the trajectory-based WCB climatology (Madonna et al., 2014), the WCBs do not need to be matched with an extratropical cyclone. The CNN models are applied to ICON output from the global 13 km domain that is remapped to 1°×1° at the relevant pressure levels.

## 3 Results

### 3.1 Climatological relationship of WCBs and extratropical cyclones

By definition, WCBs of the trajectory-based climatology by Madonna et al. (2014) are associated with extratropical cyclones. We investigate whether this relationship is found for WCBs identified with the CNN models by matching objects of WCB ascent with cyclone objects. Due to the overall highest WCB activity during Northern Hemisphere winter (DJF), results are only shown for this season.

The climatological matching frequency of trajectory-based WCB ascent and extratropical cyclones reaches more than 90%
over the western to central North Pacific and over the North Atlantic during DJF (black contours in Fig. 1a). South of the main storm track region and over continental regions values of less than 70% are found. Since the trajectory-based WCBs need to match a cyclone at least once during their 48-h life cycle, this relatively low matching frequency implies that about 30% of WCBs in these regions are matched with cyclones during their inflow or outflow stage. Though a matching criterion of WCBs and extratropical cyclones is not explicitly included in the CNN-based WCB definition, qualitatively similar results are found
(shading in Fig. 1a). In the core storm track regions the differences between the trajectory-based and CNN-based matching frequency are on the order of only –10 to 10% (shading in Fig. 1b). South of the main storm tracks the matching frequency is even higher with the CNN-based definition than with the trajectory-based definition. This suggests that the CNN models indeed identify WCBs that are associated with extratropical cyclones and not just rapidly ascending air streams which occur independently of extratropical cyclones, such as orographic ascent or convective systems. Similar results are found during
Northern Hemisphere summer (not shown). Generally, the matching frequency in summer is slightly lower than in winter, in particular over the western North Pacific. As for the winter season, the matching frequency is slightly higher when considering WCBs identified with the CNN models. Hence, we show that overall the CNN models reproduce the spatial relation of WCB ascent and extratropical cyclones.

### 3.2 Climatological relationship of WCBs and blocking anticyclones

About 10% of air masses inside Northern Hemisphere blocking anticyclones ascend by 600 hPa in 48 h during the 7 days prior to reaching the blocking anticyclone (Steinfeld and Pfahl, 2019) and thus fulfill the WCB ascent criterion. Since the WCB objects of this study are only two-dimensional objects a quantification of the air mass inside blocking anticyclones related to WCBs is not possible. Still, we estimate the matching frequency of blocking anticyclones with WCB outflows and investigate whether the same conclusions are drawn with the trajectory-based and CNN-based perspective. Here, the matching frequency is the ratio of the number of blocking anticyclones that co-occurred with WCB outflow against the number of all blocking anticyclones. During DJF, the matching frequency of trajectory-based WCB outflows and blocking anticyclones exhibits three hotspots (black contours in Fig. 2a). With matching frequencies up to 35% these are located over the western North Atlantic, the western North Pacific, and the eastern North Pacific. These areas coincide with regions with the greatest mean latent heating contribution to blocking anticyclones (Steinfeld and Pfahl, 2019). The matching frequency of blocks and WCB outflow objects identified with the CNN models exhibits a similar spatial distribution (shading in Fig. 2a). As for the Lagrangian approach, highest matching frequencies are observed over the western North Pacific, the eastern North Pacific and the western North Atlantic. The matching frequency difference between the CNN-based and trajectory-based approach is mostly on the order of –5 to 5% (shading in Fig. 2b). Largest differences exceeding 15% occur over the westernmost and eastern North Pacific. Nevertheless, as for the relation with extratropical cyclones, the CNN-based WCB outflow diagnostic reproduces the spatial relation between WCB outflow and blocking anticyclones well.

### 3.3 Application to JRA55 reanalyses

The development of the CNN models in Part I as well as applications have so far been limited to ERA-Interim data. The aim of this section is to assess whether the CNN models are also applicable to other data sets without re-training the models. To this regard, we apply the models to JRA55 data, assess their reliability against the trajectory-based WCB climatology derived from ERA-Interim, and demonstrate how the CNN models may be used to identify predictors that deteriorate their reliability. As in the previous sections the data are analysed for DJF in the testing period 01 January 2005 to 31 December 2016 and for the Northern Hemisphere.

For all three WCB stages of inflow, ascent, and outflow the reliability of the CNN models deteriorates when being applied to JRA55 reanalyses (dashed black line in Fig. 3) instead of ERA-Interim (dotted black line in Fig. 3). However, the decrease in reliability is less than 10%. Reliability curves below the gray diagonal line indicate for all stages an overestimation of the WCB probability which is most pronounced for the WCB inflow stage. As in Quinting and Grams (2021a), we perform a recalibration of the predictors by subtracting the seasonal mean difference between JRA55 reanalysis and ERA-Interim reanalysis data averaged over the period 2005–2016. For WCB inflow and outflow, the effect of the recalibration on the reliability is negligible (not shown). Only for WCB ascent the reliability of the models applied to ERA-Interim and the recalibrated JRA55 is comparable. The minor effect of a simple recalibration of the predictors on the reliability is not too surprising since CNN models learn from spatial information which remain nearly unchanged with a correction of the seasonal mean difference.





In order to identify the predictors which lead to the higher reliability in ERA-Interim than in JRA55 reanalyses, we perform sensitivity tests in which four predictors are taken from JRA55 and the fifth predictor is taken from ERA-Interim. For example for WCB inflow "JRA55 & ERAI $THA_{700}$" (Fig. 3a) means that the 700-hPa thickness advection ($THA_{700}$) is taken from ERA-

250 Interim while all other predictors are taken from JRA55 (850-hPa meridional moisture flux ($MFLY_{850}$), 1000-hPa moisture flux convergence ($MFLCON_{1000}$), 500-hPa moist PV ($MPV_{500}$), and the WCB ascent conditional probability (MIDTROP); cf. Table 1 in Part I). For WCB inflow, the reliability for the sensitivity tests "JRA55 & ERAI $THA_{700}$", "JRA55 & ERAI $MFLY_{850}$", and "JRA55 & ERAI $MFLCON_{1000}$" remains nearly the same as for "JRA55" only. However, for the sensitivity tests "JRA55 & ERAI $MPV_{500}$" and "JRA55 & ERAI MIDTROP" the reliability improves, suggesting that the 500-hPa moist

PV and the conditional probability of WCB ascent derived from JRA55 reanalyses are responsible for the reduction of the CNN models' reliability for WCB inflow.

For WCB ascent, the results are less clear due to the generally small difference between the reliability curves (Fig. 3b). It is the sensitivity test "JRA55 & ERAI $RH_{700}$" which shows the greatest improvement of the models' reliability (dashed yellow line). For WCB outflow, the 300-hPa relative humidity enhances the reliability when taken from ERA-Interim (dashed red line

in Fig. 3c). In particular, for modelled probabilities greater than 0.5 the reliability of the test "JRA55 & ERAI $RH_{300}$" nearly matches the reliability of a perfect model. The remaining predictors do not improve the reliability markedly.

Overall, in terms of their reliability the CNN models perform reasonably well on the JRA55 data set without any re-training. Also, the CNN models appear to be less sensitive to new data sets than the logistic regression models in Quinting and Grams (2021a). Moreover, the above diagnostic shows that the CNN models are not simply a black box suitable to identify footprints

of WCB inflow, ascent, and outflow but that they can also be used to identify predictors which reduce the reliability of the models. For example, instead of applying the models to different reanalysis data, they could be applied in a future study to short-range forecasts to identify erroneous predictors of WCB inflow, ascent, or outflow, which could help our understanding of the underlying interrelations and driving processes.

## 3.4 Application to operational ensemble forecasts

A major goal of the development of the CNN-based WCB diagnostic is its application to large data sets such as ensemble forecasts or climate projections. By applying the trajectory-based approach and the CNN-based approach to ECMWF's operational ensemble forecasts, we now show the effect of both approaches on the derived forecast biases in WCB occurrence frequency and on forecast skill in terms of the BSS.

### 3.4.1 WCB forecast bias

For the three winter periods analysed here and for all WCB stages of inflow, ascent, and outflow, the ensemble forecasts exhibit biases on the order of ±2.5% at a forecast lead time of 126 to 144 h (Fig. 4) for both approaches. Though biases are smaller at shorter lead times as expected, the spatial patterns shown here are also representative at lead times from 48 h onward (not shown).





For WCB inflow, the CNN-based approach reveals a dipole of positive and negative biases over the western North Atlantic (Fig. 4a). In particular, the positive biases south of the climatological WCB frequency maximum are also found with the trajectory-based approach (Fig. 4d). However, and this is the largest discrepancy between the two approaches, the negative biases to the north of the storm track regions are not identified with the trajectory approach, i.e., here the CNN based approach underestimates the WCB inflow frequency in the ensemble forecast compared to the pseudo-analysis. This discrepancy is statistically not significant in most regions as indicated by p-values larger than 0.1 of a two-sided t-test.

For WCB ascent, the biases are generally smaller than for WCB ascent. Positive biases of less than 2% occur south of the climatological frequency maximum, over eastern North America, and to the south and southeast of Greenland (Fig. 4b). Though the magnitude of the biases tends to be higher with the trajectory approach (Fig. 4e), the spatial characteristics of the biases are generally similar. Further, the differences between the two approaches are statistically not significant.

The magnitude of the WCB outflow biases locally exceed ±2.5% (Fig. 4c). Largest positive biases are found over the western North Atlantic southwest of the climatological frequency maximum as well as over the southern tip of Greenland. Negative biases occur over the central North Atlantic and Iceland. The trajectory approach yields similar mean biases that are not significantly different from the CNN-based approach (Fig. 4f). The most notable difference is that the magnitude of the negative biases tends to be smaller with the trajectory approach.

Although the differences between the two approaches are mostly not significant we briefly discuss potential reasons. First, as discussed in Part I the WCB footprints identified by the CNN models do not match the trajectory-based footprints perfectly. Second, the trajectory-based WCBs are not matched with extratropical cyclone masks (however, the CNN training data (ERA-Interim WCB climatology by (Madonna et al., 2014)) is). Accordingly, ascending air streams related to convective systems or orographic ascent may be incorrectly identified as WCBs with the trajectory approach that the CNN models are not trained to identify. Third, in contrast to the trajectory-based WCB climatology which the CNN models were trained on, the trajectories in the operational ensemble forecasts are started at a reduced number of vertical levels. Since the discrepancies between the two approaches are not significant, a deeper analysis of the possible reasons is not presented as part of this study.

### 3.4.2 WCB forecast skill

The BSS for WCB forecasts of ECMWF's operational ensemble prediction system is similar for all three WCB stages when being identified with the CNN-based approach. At initial time the BSS reaches values of 0.9 and decreases nearly linearly to values around 0.2 at 144 h lead time. A BSS of 0.2 at 144 h lead time is consistent with (Wandel et al., 2021) who found BSS values of 0.2 between 120 to 144 h lead time in ECMWF's sub-seasonal reforecast data set. For all stages, the mean BSS is lower when WCB are being identified with the trajectory approach. With the trajectory approach, the BSS reaches values of 0.7 to 0.9 at initial time and decays rapidly during the first 24 h of the forecast. A BSS of 0.0 is reached close to 144 h lead time. That the BSS is generally lower for WCBs identified with the trajectory approach than for WCBs identified with the CNN-based approach is related to the size of the WCB inflow, ascent, and outflow objects. For all three WCB stages the objects are generally larger with the CNN-based approach (not shown). Accordingly, the BSS punishes small object displacements less strongly than with the trajectory approach where the displacement of generally smaller objects leads to low BSS values.





### 3.5 Application to ICON forecasting system

This last application example focuses on the evolution of a WCB over the North Atlantic during the North Atlantic Waveguide
and Downstream Impact Experiment in 2016 (NAWDEX; Schäfler et al., 2018) from 04 October 00 UTC to 05 October
00 UTC. The main purpose is to test the applicability of the CNN models in NWP model output with considerably higher
resolution than the data we have analysed so far. At first, the synoptic evolution of the system based on ERA-Interim and
infrared satellite Gridsat clouds (Knapp et al., 2011) is depicted in Figs. 6a–c. On 4 October 00 UTC, trajectory-based WCB
inflow air parcels are located over the central North Atlantic (coloured dots in Fig. 6a). The same inflow region is also identified
with the CNN models which predict WCB inflow at a conditional probability greater than 0.4 (red contours in Fig. 6a). In the
following 12 hours, the WCB air parcels ascend northeastward ahead of the cold front and reach the mid troposphere (coloured
dots in Fig. 6b). Most ascending air parcels are found in the northern half of the cyclone along the warm front (not shown)
and only a few trajectories ascend directly ahead of the cold front. Still, both areas of ascending trajectories are depicted by
the CNN model (green contours in Fig. 6b). On 5 October 00 UTC, the WCB outflow is characterized by a broad cloud shield
extending from the southern tip of Greenland to Iceland (Fig. 6c). The air parcels indicate the characteristic cyclonic and
anticyclonic branches of the WCB (see Martínez-Alvarado et al., 2014). Both branches are identified by the CNN model which
predicts the occurrence of WCB outflow at a conditional probability greater than 0.4 (blue contours in Fig. 6c).

The ICON simulation initialized on 03 October 2016 00 UTC depicts the WCB evolution compared to ERA-Interim reason-
ably well (Fig. 6). On 04 October 00 UTC, a large number of trajectories is located in the WCB inflow layer over the North
Atlantic (dots in Fig. 6d). It should be noted that the trajectories in the ICON simulation are started hourly and at a spatial
grid spacing of 25 km explaining the larger number of trajectories than in ERA-Interim where trajectories are started 6-hourly
at a grid spacing of 80 km. Moreover, trajectories calculated from high-resolution model output tend to be characterized by
average ascent rates of 600 hPa in considerably less than 48 h (e.g., Oertel et al., 2020). Hence, displaying 48-h WCB air parcel
positions results in a large number of trajectories that are prior to their coherent ascent or have already finished their ascent for
several hours (see light grey colored dots in Fig. 6d,f), and thus appear spread out in the cyclones warm sector or recirculate
in the upper-tropospheric ridge. Focusing only on those air parcels that are about to ascend to the ascent layer within the next
6 hours (dark grey and coloured dots in Fig. 6d), the inflow region is spatially more confined and in the same region as in
ERA-Interim. This inflow region is well depicted by the CNN model which predicts inflow at a conditional probability greater
than 0.4.

The ascending WCB air parcels on 04 October 12 UTC are shown in Fig. 6e. As in ERA-Interim ascending air parcels
are found along the warm front and immediately ahead of the cold front (not shown). However, the number of air parcels
ascending ahead of the cold front is considerably larger than in ERA-Interim. This is likely due to faster ascent and resolved
convection in the ICON simulation which explicitly resolves convective ascents instead of parameterizing them as in ERA-
Interim. Regions of WCB ascent as predicted by the CNN model are nearly collocated with the ascending air parcels identified
with the trajectory-based approach. This collocation is quite remarkable keeping in mind that the CNN model was trained on



ERA-Interim with coarser spatial resolution and that the trajectories are calculated from data at high resolution in the refined nest.

On 05 October 00 UTC, the CNN models predict the occurrence of WCB outflow in a region extending from the southern tip of Greenland to Iceland (Fig. 6f). This outflow region is collocated with the outflow identified in ERA-Interim by both the trajectories and the CNN model. In the ICON simulation, however, a significant fraction of trajectories reaches the outflow layer further south and ahead of the cold front. This outflow, which is related to rapid and partly convectively driven ascents directly ahead of the cold front, is not captured by the CNN model which highlights one limitation of the CNN when being trained on ERA-Interim but applied to convection permitting simulations. Due to the 24 hour time-lag between the conditional probability of ascent and the actual WCB outflow, the CNN model is trained to capture slantwise ascending air masses with relatively low ascent rates, such as in ERA-Interim, but not convective rapid ascents. Accordingly, the northern most part of the outflow is depicted reasonably well in the ICON simulation as well as in ERA-Interim. However, the southern part of the WCB outflow in the ICON simulation which is related to rapid ascents along the cold front, is not captured by the CNN model. Thus, we hypothesize that the time-lag of 24 hours between the conditional probability of WCB ascent as predictor and the actual outflow is a too strong constraint when applying the CNN model to convection permitting simulations often characterized by WCB ascent timescales of less than 48 hours. This is confirmed when applying a CNN which does not use the conditional probability of ascent as predictor (referred to as *standard model* in Part I). Rather, it uses information from the four physical predictors which are 300-hPa relative humidity, 300-hPa divergent wind speed, 500-hPa static stability, and 300-hPa relative vorticity, and the running-mean WCB climatology. With these predictors the WCB outflow as predicted by the CNN extends further southward (dashed blue contours in Fig. 6f) and captures large-parts of the outflow based on the trajectory approach.

## 4 Conclusions and Outlook

In Part I of this two-part study, we introduced novel CNN-based models that skillfully identify footprints of WCB inflow, ascent, and outflow from data at a comparably coarse temporal and spatial resolution which would not be suitable for trajectory calculations. With the CNN-based models we are now capable of evaluating the representation of WCBs in large data sets such as ensemble forecasts or climate projections at comparably low computational costs. The present Part II shows the versatile applicability of the CNN-models to different data sets such as reanalysis, ensemble forecasts, and convection permitting simulations and compares the results with the trajectory-based counterpart.

The application of the CNN-based models to ERA-Interim reanalysis data and the matching of WCB objects with extratropical cyclone and blocking objects identifies two well-known relationships.

1. The ascent of WCBs is associated with and contributes to the intensification of extratropical cyclones (Madonna et al., 2014; Binder et al., 2016). With the trajectory approach and the CNN-based approach it is found that in the main storm-track regions up to 90% of WCB ascent objects co-occur with an extratropical cyclone object. Though a matching criterion of WCBs and extratropical cylones is not explicitly included in the CNN-based WCB definition compared to the trajectory-based definition Madonna et al. (2014), quantitatively similar results are found with either approach.





This suggests that the CNN models indeed identify air streams that are associated with extratropical cyclones and not
just rapidly ascending air streams which occur independently of extratropical cyclones such as orographic ascent or
convective systems.

2. About 10% of air masses in Northern Hemisphere blocking anticyclones are related to WCBs (Steinfeld and Pfahl, 2019).
   Due to the two-dimensional nature of the WCB objects, the proportion of WCB air mass in blocking anticyclones cannot
   be quantified with the CNN-based approach. Still, we find that locally up to 35% of blocking anticyclones co-occur with
   WCB outflow. Interestingly, the areas of highest matching frequency coincide with regions with the greatest mean latent
   heating contribution to blocking anticyclones (Steinfeld and Pfahl, 2019).

Future studies could use the CNN models to perform analyses alike in climate model projections. To the authors' knowledge a
systematic investigation of the matching frequency of cyclones, blocking anticyclones, and WCBs has not been conducted yet
in these data sets.

When being applied to data sets other than the CNN models were trained on, the reliability of the models deteriorates slightly.
Still, this information is shown to be useful to identify predictor fields that cause the deterioration. Similar to the application to
JRA-55 reanalyses data as in this study, future studies could apply the CNN models to short-range forecasts in order to identify
those predictors that cause the difference in reliability compared to ERA-Interim. Such an approach could be useful to identify
NWP or climate model biases in basic atmospheric variables and which would help improving the model representation of
WCBs specifically and model improvement in general in the long term.

The application of the CNN models to operational ensemble forecasts reveals biases in the WCB occurrence frequency over
the North Atlantic. Though the period considered here includes only three winter seasons, the overestimation of WCB inflow,
ascent, and outflow to the south of the climatological WCB frequency maximum and an underestimation of WCB outflow in the
North Atlantic is consistent with (Wandel et al., 2021). That the trajectory and CNN-based approach identify similar biases in
operational ensemble forecasts encourages us to use the CNN models to systematically investigate the representation of WCBs
in large data sets. Future studies could apply the diagnostic in inter-model comparisons on weather to climate time-scales. Data
sets such as the THORPEX Interactive Grand Global Ensemble (TIGGE Swinbank et al., 2016), the subseasonal to seasonal
prediction project data base (Vitart et al., 2017), or the Coupled Model Intercomparison Project Phase 6 (CMIP6 Eyring et al.,
2016) provide numerous opportunities.

Finally, we would like to stress that the CNN models introduced in this study are limited in the sense that they only provide
information about the occurrence of WCB inflow, ascent, and outflow. Thus, the models are optimally suited to be applied
to large data sets. For process-oriented studies on the physical properties of WCBs the trajectory approach yields invaluable
insights and should thus be preferred. Future developments of CNN-based WCB diagnostics that account for the associated
mass transport or the three-dimensional spatial and temporal evolution could provide additional insights and an even more
accurate identification of WCBs. Our study illustrates that deep learning methods can be used efficiently to support process-
oriented understanding of forecast error and model biases and opens numerous new directions for NWP and climate model
verification and process-oriented research in large data sets.



*Code and data availability.* The exact version of the *time-lag models*, the decision thresholds, the 30-d running-mean trajectory-based WCB climatology, code to process the input data for the models as well as post-processing scripts to generate the figures of this paper are pro-
vided via the following repository https://git.scc.kit.edu/nk2448/wcbmetric_v2.git and archived on Zenodo (https://doi.org/10.5281/zenodo.5154980). ERA-Interim data are freely available at https://apps.ecmwf.int/datasets/data/interim-full-daily. Monthly ERA-Interim based climatologies of extratropical cyclones and blocks can be downloaded at http://eraiclim.ethz.ch/. JRA-55 data were retrieved from https://doi.org/10.5065/D6HH6H41. The LAGRANTO source code and documentation can be downloaded from http://www.lagranto.ethz.ch. The ICON source code is distributed under an institutional license issued by the German Weather Service (DWD). For more information see
https://code.mpimet.mpg.de/projects/iconpublic. The model output of the ICON simulation is available from the authors upon request.

*Author contributions.* JQ, AO, and MP performed the data analysis. CMG set up and curated the real-time data archive. All authors contributed to the interpretation of the data and the preparation of the paper.

*Competing interests.* The authors declare that they have no conflict of interest.

*Acknowledgements.* This work was funded by the Helmholtz Association as part of the Young Investigator Group "Sub-seasonal Predictabil-
ity: Understanding the Role of Diabatic Outflow" (SPREADOUT, grant VH-NG-1243). The research was partially embedded in the subprojects A8 and B8 of the Transregional Collaborative Research Center SFB/TRR 165 'Waves to Weather' (https://www.wavestoweather.de) funded by the German Research Foundation (DFG). The high-resolution ICON simulation was performed on the supercomputer ForHLR II funded by the Ministry of Science, Research and the Arts Baden-Württemberg, Germany, and by the German Federal Ministry of Education and Research. Sincerest thanks to the Atmospheric Dynamics group at ETH Zurich in particular to Michael Sprenger and Heini Wernli for
sharing the trajectory-based WCB data, the extratropical cyclone data, and the blocking anticyclone data. ECMWF, Deutscher Wetterdient, and MeteoSwiss are acknowledged for granting access to the ERA-Interim data set and operational ensemble forecast data.



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

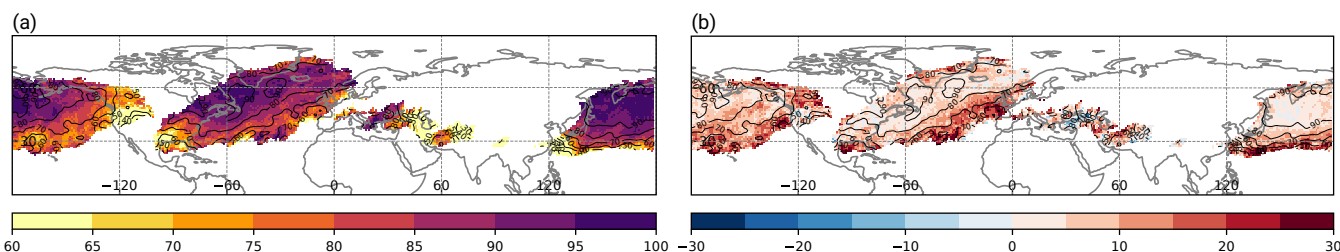

**Figure 1.** (a) Climatological matching frequency of WCB ascent footprints with extratropical cyclones during DJF as identified with the trajectory-based approach (black contours in %) and the CNN-based approach (shading in %). (b) Contours as in (a), but shading denotes the difference in matching frequency (in %) between the CNN-based and trajectory-based approach.





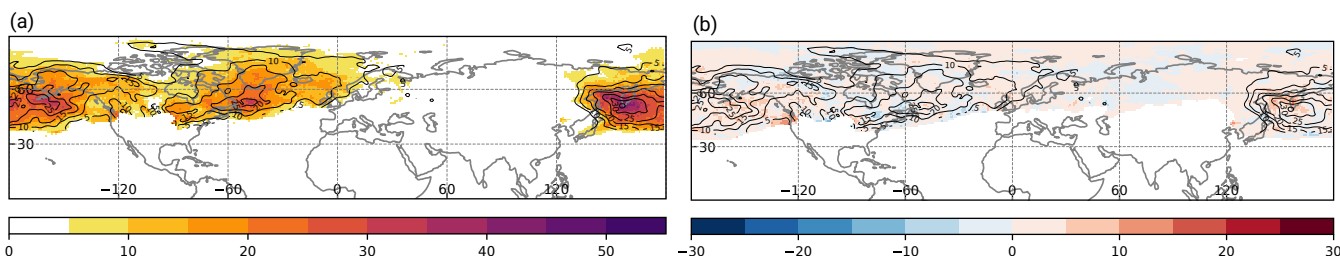

**Figure 2.** (a) Climatological matching frequency of blocking anticyclones and WCB outflow footprints during DJF as identified with the trajectory-based approach (black contours in %) and the CNN-based approach (shading in %). (b) Contours as in (a), but shading denotes the difference in matching frequency (in %) between the CNN-based and trajectory-based approach.



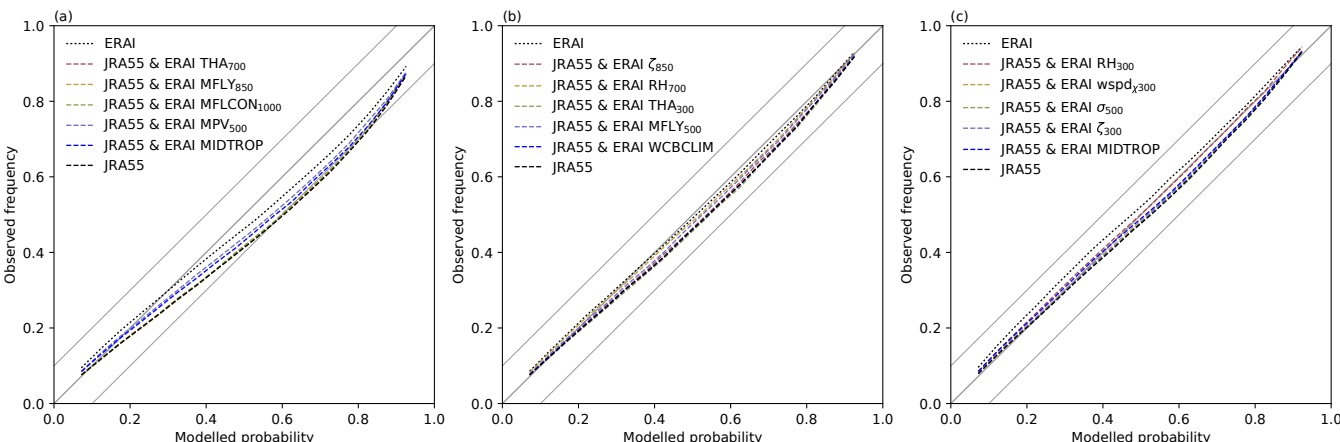

**Figure 3.** Reliability diagrams of the CNN models for (a) WCB inflow, (b) WCB ascent, and (c) WCB outflow when being applied to ERA-Interim (dotted black line) and JRA55 (dashed black line). Coloured dashed lines show the reliability for sensitivity experiments outlined in the main text. Please see Table 1 in Part I for the abbreviations of predictors. Probabilities modelled with the CNN models (x-axis) and observed frequencies from the trajectory-based data set (y-axis) are binned into 19 bins based on the modeled probabilities. The perfect modeled probability and a $\pm 10\%$ interval about the perfect model is shown by the solid and dashed diagonals, respectively.



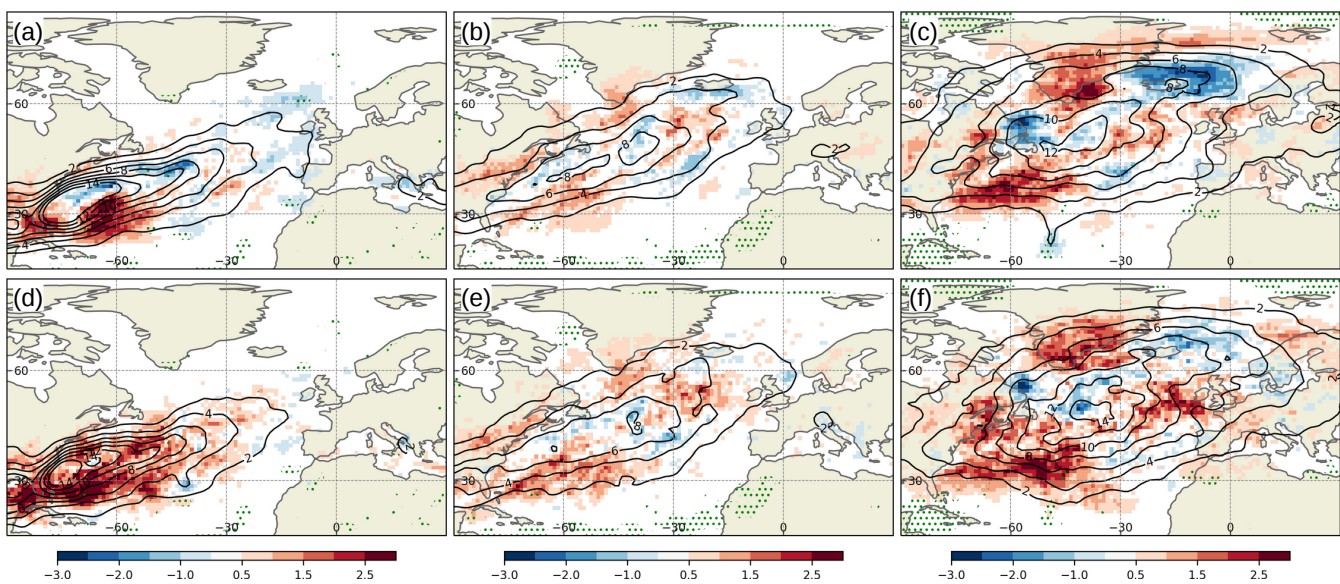

**Figure 4.** WCB frequency biases (shading in %) in ECMWF's operational ensemble forecasts averaged over forecast leadtime 126 to 144 h as diagnosed with the (a, b, c) CNN-based approach and (d, e, f) with the trajectory-based approach. (a, d) show WCB inflow, (b, e) WCB ascent, and (c, f) WCB outflow. Black contours denote the WCB frequency (%) in the period 01 December to 28 February in the three winters 2018/19, 2019/20, and 2020/21 for the respective WCB stage. Areas where the biases identified with the two approaches are significantly different (p-value < 0.1) are highlight by green dots.





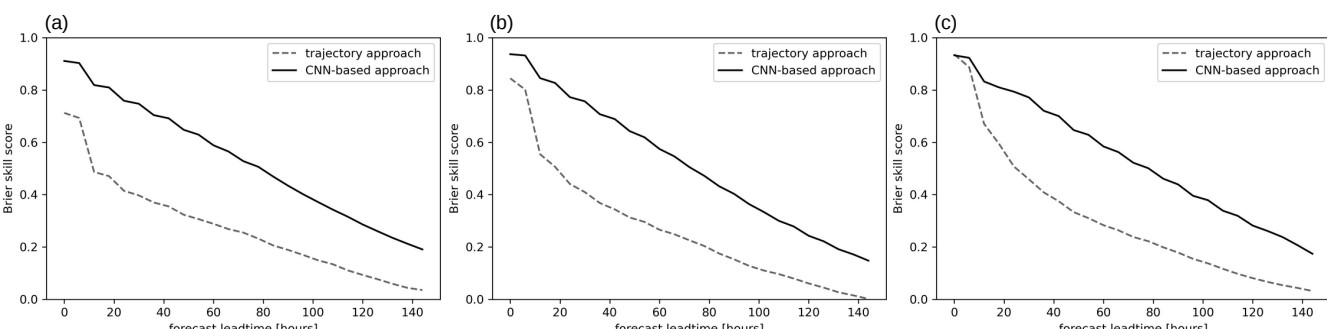

**Figure 5.** Mean Brier skill score for WCB (a) inflow, (b) ascent, and (c) outflow forecasts of ECMWF's operational ensemble forecasts averaged over the region shown in Fig. 4.



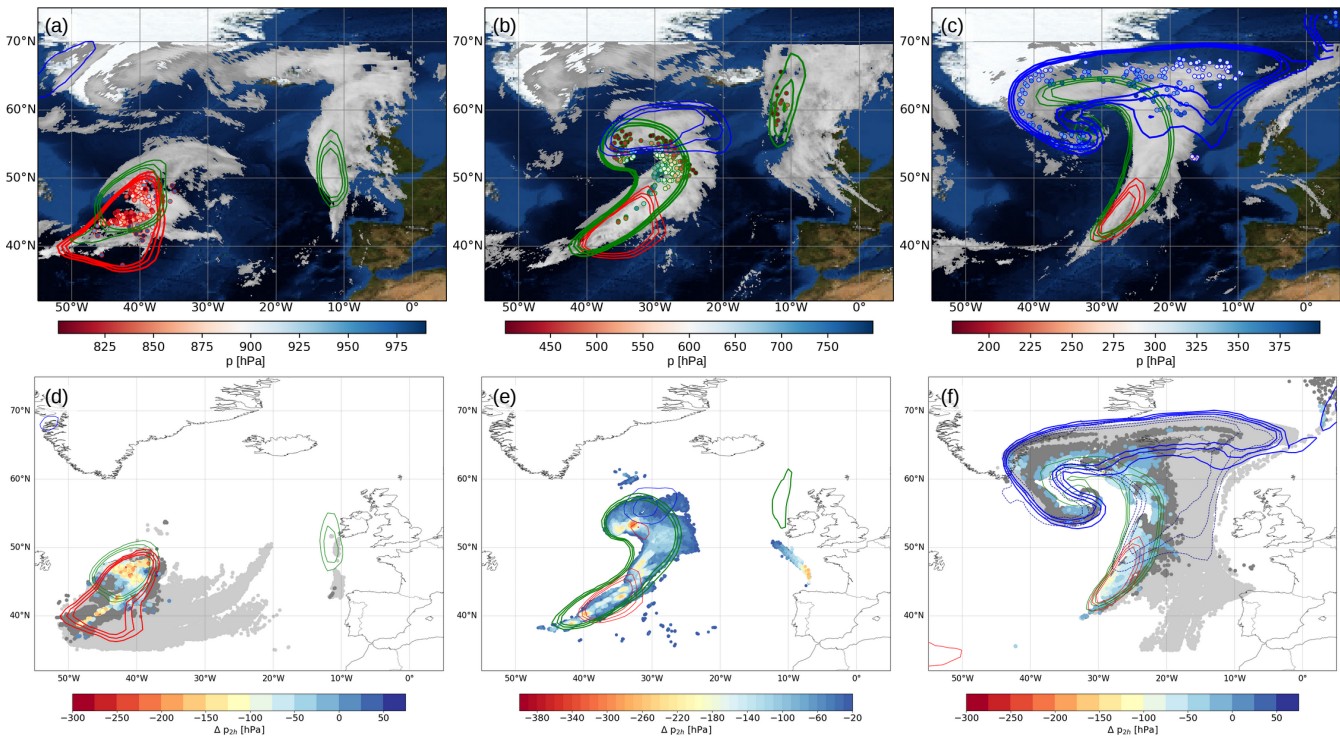

**Figure 6.** Synoptic evolution of WCB case study on (a,d) 04 October 2016 00 UTC, (b,e) 04 October 2016 12 UTC, and (c,f) 05 October 2016 00 UTC. Panels (a–c) are based on ERA-Interim and show conditional probability of WCB inflow (red contours), ascent (green contours), and outflow (blue contours) all at 0.2, 0.3, 0.4. Dots show locations of trajectory-based WCB air parcels for (a) inflow, (b) ascent, and (c) outflow and are coloured according to their pressure (hPa). Further are shown IR Gridsat clouds (Knapp et al., 2011) with a brightness temperature of less than –10°C. Panels (d–f) are based on convection permitting simulations with the ICON model. Contours are the same as in (a–c) except for (f) where the dashed blue contour denotes the conditional WCB outflow probability calculated with the *standard model* of Part I. Dots show trajectory-based WCB air parcels for (d) inflow, (e) ascent, and (f) outflow that will transition to the ascent layer in the next hour (colored dot), or have just arrived in the outflow layer, respectively, (colour indicates 2-h pressure change), that have been in the layer for 1–6 hours (dark gray dots), and those that have been in the corresponding layer for more than 6 hours (light gray dots).