# Peer review of "EuLerian Identification of Ascending air Streams (ELIAS 2.0) in Numerical Weather Prediction and Climate Models. Part II: Model application to different data sets"

_Geoscientific Model Development, 2021_

## Author Response (AR1)

**Reviewer 1**

In this study, the authors demonstrated the application of a new CNN-based WCB identification method in different datasets and compared it with the trajectory-based method. The new CNN-based method introduced in this paper can obtain comparable results with the trajectory-based method, while the trajectory-based method requires higher spatial and temporal resolution, as well as more expensive computer power. It could be a powerful high-efficiency process-oriented tool for multiple purposes. The authors did a great job in the analysis and comparisons. Overall, I think this manuscript is publishable in GMD.

Dear Reviewer,

We are very grateful for your positive feedback and the detailed comments which helped to improve the clarity of the manuscript. In the following, we respond point by point to your comments. Our responses are highlighted in blue. For your convenience, we have also uploaded a track-changes version of the manuscript.

Kind regards,
Julian Quinting, Christian Grams, Annika Oertel, & Moritz Pickl

**Specific Comments**

(1) Line 90 "For WCB ascent, a fifth predictor is the 30-day running mean trajectory-based climatological WCB occurrence frequency." Does it mean that for a specific dataset or model if you want to use the CNN-based approach, you need to run the trajectory-based approach first to obtain the 30-day running mean climatological WCB occurrence frequency?

Thank you for this comment. If one wanted to apply the CNN-based approach to a specific dataset a calculation of the trajectories is not necessary. As part of the gitlab repository we provide the ERA-Interim-based 30-day running mean climatology of WCB inflow, ascent, and outflow (https://git.scc.kit.edu/nk2448/wcbmetric_v2/-/tree/master/data). Since the WCB climatology is the least important predictor for the CNN, this indicates an overall small sensitivity to changes in the WCB climatology. Accordingly, we are convinced that the CNN-based approach will also be applicable to climate model projections in which the overall WCB climatology might change. In the revised manuscript, we now include the information that the climatology is available from the gitlab repository (l. 95): „A fifth predictor is the 30-day running mean trajectory-based climatological WCB occurrence frequency which is provided via the corresponding GitLab repository and thus does not need to be calculated prior to using the models (see Code and data availability section)".

(2) Line 103-104 "Here, we test for this relationship by matching the trajectory-based and CNN-based masks of WCB ascent with the extratropical cyclone masks …". In my understanding, for the relationship between cyclones and WCBs, "matching" in this study is defined using only the WCB ascent, while the "matching" in the trajectory-based method (line 76-78) is defined using all WCB objects, including inflow, ascent, and outflow. That is confusing. Why did not the authors use a consistent definition of "matching" between WCBs and cyclones?

You are right. In the original trajectory-based WCB definition of Madonna et al. 2014, „matching" is defined using all WCB objects. However, in this study we consistently match only WCB ascent with the cyclone mask for both the trajectory- **and** CNN-based approaches. This information is provided in line 113 of the manuscript. We chose the WCB ascent for matching since this is dynamically most strongly related to extratropical cyclones (e.g., Binder et al. 2016). In the revised manuscript, we now include the reference to the study by Binder et al. 2016 (l. 115).

(3) At lines 294-301, the authors discussed the potential reasons for the differences between the two approaches nicely. But do the authors have any ideas why the differences for WCB inflow and outflow are larger than the difference for WCB ascend in Figure 4?

The way the results are shown in Fig. 4 is possibly misleading in this regard. In terms of absolute numbers we agree that the differences for WCB inflow and outflow are larger than the difference for WCB ascent. However, the mean absolute occurrence frequency of WCB ascent is smaller than the occurrence frequency of inflow and outflow. Accordingly, relative to the mean occurrence frequency the differences for ascent are similar to those for inflow and outflow. This explanation is provided as a footnote in the manuscript (bottom of page 10): „One should note that the mean absolute occurrence frequency for WCB ascent is lower than for WCB inflow so that the biases relative to the mean occurrence frequency are similar to the relative biases of WCB inflow".

(4) At lines 310-311 "For all three WCB stages the objects are generally larger with the CNN-based approach (not shown)." This is interesting. What is the difference (%) of the WCB objects between the CNN-based approach and the trajectory-based approach on average?

WCB ascent objects identified with the trajectory-based approach only have 60% of the size identified with the CNN-based approach. This is due to a considerably higher number of objects smaller than $0.2*10^6$ km$^2$ when applying the trajectory approach (Fig. 1 of this document). So, objects which are larger with the CNN-based diagnostic are split into several small objects with the trajectory approach. If we only evaluate objects larger than $0.2*10^6$ km$^2$ the difference reduces to 10%. We provide this explanation in the revised version of the manuscript (l. 320): „For all three WCB stages the WCB objects identified with the trajectory-based approach only have approximately 60% of the size of objects identified with the CNN-based approach (not shown). For WCB ascent, for example, this is due to a very large number of objects smaller than $0.2×10^6$ km$^2$ when applying the trajectory approach."

[Figure]

*Fig. 1. Distribution of the object size of WCB ascent objects identified with the CNN-based approach (left) and with the trajectory-based approach (right).*

(5) The difference of BSS in Figure 5 is relatively large. In addition to the difference of WCB sizes as described at lines 310-312, are there any other reasons?

Thank you for this thought-provoking question. We discussed it among the co-authors with the conclusion that the main reason for the difference of BSS for the trajectory vs. the CNN approach is the size of the WCB objects.

(6) Line 401-404, these datasets have quite different horizontal resolution. Is the CNN-based approach sensitive to the horizontal resolution? In this study, all datasets were remapped to 1x1 degree grid. Is that a necessary step for the CNN-based approach?

The CNN-based approach is sensitive to the horizontal resolution in the sense that the input data need to be provided on 1x1 degree grid. Still, as we show with the application to convection permitting simulations, the native resolution of input data before remapping can in principle be much higher. We now include this important information in Section 2.1.2 of the revised manuscript by explaining that „a mandatory step before applying the CNN models is to remap the predictors to a regular 1°×1° latitude-longitude grid" (l. 102).

**Technical Corrections**

(1) Line 28-29 "on the order of 20K", is that 20K through the WCB life cycle or per day?

This was indeed not clear. It is 20 K through the WCB life cycle, i.e., 20 K in 48 hours. We added this information (l. 29) and now state „This WCB ascent, which can be slantwise or convective in nature (e.g.,Neiman and Shapiro, 1993; Rasp et al., 2016; Oertel et al., 2019), is accompanied by latent heat release on the order of 20 K during its life cycle due to phase changes during cloud formation (Eckhardt et al., 2004; Madonna et al., 2014)".

(2) Line 77 "an extratropical cyclone mask", please define it when it appears for the first time? Does it have a consistent definition through this manuscript?

Thanks for this suggestion. We include the definition of the cyclone mask in the revised manuscript when it is first mentioned (l. 78). Following Wernli and Schwierz (2006), cyclone mask is defined as the region enclosed by the outermost closed sea level pressure contour of a cyclone. This definition is used consistently through the manuscript.

(3) Line 115 "masks of blocks", similar with the last comment, please define "masks of blocks".

Likewise we now include a definition for the masks of blocks. Following Pfahl et al. (2015), the blocking mask is defined by grid points where the vertically integrated PV is 1.3 pvu lower than the monthly climatology for at least five consecutive days (l. 125).

(4) Line 129-130, please clarify what are "CY45r1, CY46r1, and CY47r1".

The acronyms CY45r1, CY46r1, and CY47r1 are used as identifiers at ECMWF to specify their model cycles. In the revised manuscript, we now include information on the period when the respective cycles were used. This is 5 June 2018 to 10 June 2019 for CY45r1, 11 June 2019 to 29 June 2020 for CY4r1, and 30 June 2020 to 10 May 2021 for CY47r1 (l. 136).

(5) Line 197 "Due to the overall highest WCB activity during Northern Hemisphere winter (DJF)", please add a reference here.

We now include a reference to the Madonna et al. (2014) paper where this information is provided (l. 205).

(6) Line 285 "For WCB ascent, the biases are generally smaller than for WCB ascent." Do the authors mean "… smaller than for WCB inflow"?

Thanks a lot for spotting this error. Yes, we actually meant „smaller than for WCB inflow". We corrected the manuscript accordingly and now state in l. 293 „For WCB ascent, the biases are generally smaller than for WCB inflow".

**Reviewer 2**

This manuscript presents the exemplary application of the deep-learning-based identification methods for WCB. The method itself is shown to be computation time and data saved, and thus it is should be useful for large data analyses such as for large simulations from regional and climate models, as well as the weather forecast ensembles. Therefore, I think this manuscript is acceptable after some minor revise as follows.

Dear Reviewer,

We are very grateful for your positive feedback and the valuable comments. In the following, we respond point by point to your comments. Our responses are highlighted in blue. For your convenience, we have also uploaded a track-changes version of the manuscript.

Kind regards,

Julian Quinting, Christian Grams, Annika Oertel, and Moritz Pickl

(1) This manuscript is the second part of the series of the proposed deep-learning-based identification methods for WCB, so I have to go through the first part before going to this method application part. For one standalone piece, I suggest the authors to breifly describe the deep-learning method and the predictors and predictant. In the current manuscript, it is impossible to fully understand why the authors adopted the five preditors, especially there is a preditor (Line 90) from the lagrangian analyses as I understand from the first part? In my understanding, the method can improve the efficiency in WCB identification but cannot work without previous trajectory analyses, which could limit the deep-learning application.

We very much appreciate these suggestions. In the revised manuscript, we include additional information on the deep-learning method and in particular name the individual predictors (l. 93ff). We very much hope that the additional information will help to get the essential information. It is correct that the WCB climatology derived with the trajectory approach is needed. However, we provide this data set as part of the repository so that the diagnostic can directly be applied with out recalculating the trajectory data set. To clarify this aspect, we now state „A fifth predictor is the 30-day running mean trajectory-based climatological WCB occurrence frequency which is provided via the corresponding repository and thus does not need to be calculated prior to using the models (see Code and data availability section)" (l. 94).

(2) In the first part, the authors cite the following one: Quinting, J. F., Grams, C. M., and Oertel, A.: Deep learning for the Verification of Warm Conveyor Belts in NWP and Climate Models. Part II: Model application, Weather and Climate Dynamics, pp. 1–66, 2021. Did the authors submit this manuscript to Weather and Climate Dynamics before? If yes, do the authors get some comments and make some improvements.

Thank you very much for spotting this error. The reference in Part I was incorrect. We have not submitted this manuscript to Weather and Climate Dynamics before. Actually, this was our original

idea which we then discarded. In the revised version of Part I, we corrected the reference and now refer to this paper in GMD.

(3) Some sentences should be corrected. For example, "South of the main storm track region and over continental regions values of less than 70% are found. " it is not clearly which value?

Thank you very much for pointing this out. We had a careful read of the manuscript and corrected the sentences where necessary. For the sentence above we now state „South of the main storm track region and over continental regions the matching frequency is locally less than 70%" (l. 209). For all changes, we refer to the track-changes version of the manuscript.